# Aircraft turnaround time dynamic prediction based on Time Transition Petri Net

**Yanyu Cui** [1], **Linyan Ma**[2], **Qingmiao Ding** [1]*, **Xuan He**[3], **Fanghui Xiao**[4], **Bin Cheng**[2]

**1** College of Aeronautical Engineering, Civil Aviation University of China, Tianjin, China, **2** College of Transportation Science and Engineering, Civil Aviation University of China, Tianjin, China, **3** The Second Research Institute of CAAC, Chengdu, Sichuan, China, **4** Shenzhen Airport (Group) Co. Ltd., Shenzhen, Guangzhou, China

* qmding@cauc.edu.cn

## Abstract

Accurate aircraft turnaround time prediction is an important way to coordinate the operation time of airport ground service and improve the efficiency of airport operation. In this paper, by analyzing the aircraft turnaround operation process, a description model based on Time Transition Petri Net is proposed. The model describes the flight turnaround operation process and the logical relationship of the operation. According to the model, a dynamic prediction method of turnaround time based on Bayesian theorem is designed. According to the actual landing time of the flight, the aircraft turnaround time is predicted. The specific method is to obtain the prior probability distribution and joint distribution law of each operation link according to the flight history data, and use Shapiro-Wilke to test the prior probability distribution of each link. Based on the analysis and comparison between the actual turnaround data of a large airport in China and the forecast data proposed in this paper, the root-mean-square error 3.75 minutes and the mean absolute error 3.40 minutes can be calculated. This paper contributes to the improvement of flight punctuality rate and airport clearance level.

## 1. Introduction

As of 2023, the total number of civil aviation airports in China reached 259, with a total capacity of 1.56 billion passengers. There are 20 airports have a passenger throughput of more than 20 million. The rapid development of such airports makes the number of inbound and outbound flights continue to rise, increasing the pressure of airport flights passing the station. To improve the operation efficiency of the airport, the operation management level of the airport needs to be further improved. The aircraft turnaround operation is carried out by a series of logical operation links as the core. With the continuous development and improvement of the existing A-CDM operation control system, aircraft turnaround time prediction can coordinate the operation time of the airport ground service and improve the airport operation level.

Due to the influence of various factors [1–3], it is very difficult to accurately predict the aircraft turnaround time. Moreover, the research results have distinct characteristics of airport data samples, and the portability of analysis results is low. Bin Yu [3] believed that few papers

**Funding:** This research was funded by the National Key Technologies Research and Development

Program of China, grant number
2021YFB1600500-2021YFB1600502. The funders
had no role in study design, data collection and
analysis, decision to publish, or preparation of the
manuscript.

**Competing interests:** The authors have declared
that no competing interests exist.

have studied the causes of flight delay from a microscopic perspective (weather, season, delay propagation). He mined, identified and measured the importance of key micro-factors affecting flight delays from the perspective of route conditions, airport reservation and actual congestion. In most studies, Petri net [4–6], Bayesian network [7–9], neural network [10–12] and other methods were used to complete the prediction. Xing [5] proposed a quantitative description model of flight ground support service based on Colored Petri Nets (CPN) network, used Monte Carlo method for time estimation, and later [10] established a Gaussian probability model of arrival time of flight ground service vehicles. The results show that the prediction accuracy of this method is 3%-5% higher than that of traditional BP neural network and traditional Bayesian network. Wang [7] made a detailed prediction of each support environment, and constructed a dynamic prediction method of ground support process based on the traditional Bayesian network combined with Gaussian kernel probability density estimation. It is verified that the proposed method is better than the deep neural network. The research of Sanchez [13] not only had the performance of detail prediction, but also explored the influence of the interaction in the steps and the delay of different links on the total support time, reduced the propagation of delay, enhanced the robustness in the process of turnaround operation, and constructed a bridge from the dynamic prediction of ground support time to the optimization of ground service time. In terms of application, many studies are based on the prediction of aircraft turnaround time. Xu [14] designed an algorithm to improve the adaptability of vehicle scheduling to uncertain operation time based on the historical data of an airport in China. Andreatta [15, 16] designed a ground operation vehicle allocation sequence program, which is combined with the real-time information of the apron network provided in the AAS platform to allocate the ground operation vehicles to each task according to the flight take-off sequence. Compared with the traditional manual scheduling, the intelligent program has the advantages of fast allocation speed and short vehicle driving distance. Fitouri-trabelsi [17] proposed a decentralized management method for the problem of multiple vehicles on the ground, strengthened the information transmission of ground dispatchers, ground handling sub-managers, and ground operators, and allocated necessary vehicles and other resources near the departure or arrival gates to reduce the amount of calculation and a large amount of information transmission.

Under the background of increasingly busy airports, it is of great significance to provide passengers with reliable travel schedules and improve the service performance of airports and airlines. Petri Nets are used in the analysis of operation processes in various industries, such as logistics warehousing, manufacturing and assembly, signal communication. It has a good ability to describe discrete events. This paper first sorted out the necessary support activities of the flight turnaround through the station, thus establishing the Time Transition Petri Net (TTPN), and then combined Bayesian theorem to predict the flight turnaround time, analyzed and verified the prediction model with actual data, and finally proved that the prediction method has certain reference value.

## 2. The turnaround process description

The turnaround process refers to a series of ground operations in the flight area between the landing of the aircraft and the take-off of the next flight task, including a series of ground service work for passengers, luggage and aircraft, providing the resources and ground services required for the aircraft to perform the next flight normally, usually in parallel or in sequence. This article mainly takes the operational data of a hub airport in China in 2023 as an example, and selects commonly used aircraft models such as A320 and B777. The ground operation process is roughly shown in Fig 1.

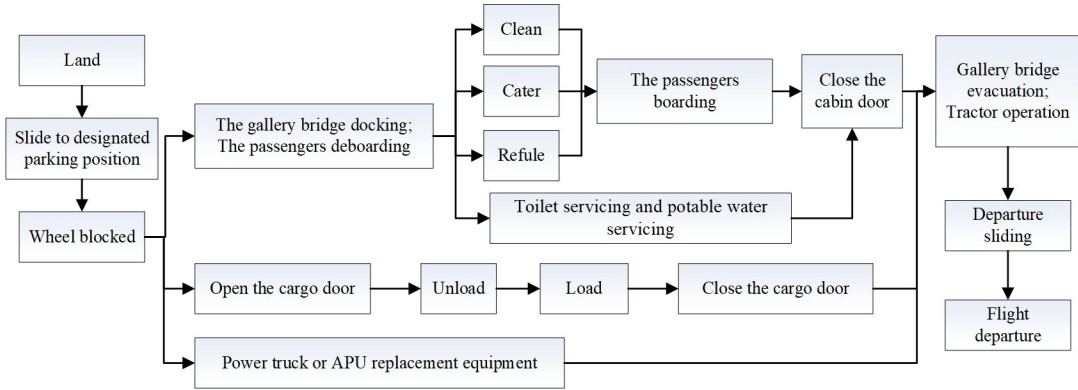

**Fig 1. Flight turnaround operation process.**

## 2.1 Time Transition Petri Net (TTPN)

The Petri net was proposed by German scientist Carl Adam Petri in his doctoral thesis in 1962. This mesh model can describe the discrete event system well and express the concept of concurrency more intuitively. However, due to the lack of characterization of the transition time factors, it cannot meet the modeling requirements of this paper. In view of this, this paper uses the TTPN model to describe the turnaround operation process.

TTPN extends the basic Petri net to six elements: $(PN, \delta) = (P, T, \delta, Pre, Post, M_0)$. $PN$ is the basic form of Petri net. $P = \{p_1, p_2, \ldots, p_n\}$ is a finite set of places. $\{p_1, p_2, \ldots, p_n\}$ represent the status of different turnaround operations respectively. $T = \{t_1, t_2, \ldots, t_n\}$ is a set of finite transitions. $\{t_1, t_2, \ldots, t_n\}$ represent different turnaround operation tasks respectively. $\delta = (\delta_1, \delta_2, \ldots, \delta_n)$ is a set of all transition's durations. $Pre$ is the weight function of the input place to the transition. $Pre(p_0, t_0)$ represents from $p_0$ to $t_0$ weight value. $Post$ is the weight function of the transition to the output place. $Post(p_0, t_0)$ represents from $t_0$ to $p_0$ weight value. In this paper, the weight function is 1, which means that each step needs to consume a resource. The resource is the token in the petri network, and the flow of token can intuitively describe the flight over the station. $M$ represents the state of the TTPN model, the number of elements represents the number of tokens in the corresponding numbered place. $M_0$ represents the initial state of TTPN. In the network, the circle is used to represent the place, the rectangle is used to represent the transition, the directed arc is used to represent the relationship between the place and the transition, and the black dot is used to represent the Token.

Each turnaround operation is regarded as a component element in TTPN, and the process is described with relevant basic elements. $T = \{t_1, t_2, \ldots, t_n\}$ represents $n$ turnaround operations. $\delta: T \to R^+$ represents the duration of all operational services. If conditions are met, $\forall p_i \in t_i, M(p_i) \geq Pre(p_i, t_i)$, then transition $t_i$ is said to be enabled, have the right to occur. Denoted as $M[t_i>$. Once the transition $t_i$ is enabled, the corresponding token in the weight function is immediately removed from each input place of the transition. Transition excitation after delay time $\delta_i$, and output the token to its output place. The precondition is satisfied, and the occurrence of the event changes the local state related to the event at the same time, and the occurrence of the event is described by the excitation of the enabling transition. If in state $M$, the excitation of enabling transitions will create new states, then denoted as $M'$. The process is recorded as $M[t_i>M'$. $M_0 = (1,0,0\ldots0)$, $M'$:

$$\forall p \in P : M' = M(P) - Pre(p, t) + Post(p, t) \tag{1}$$

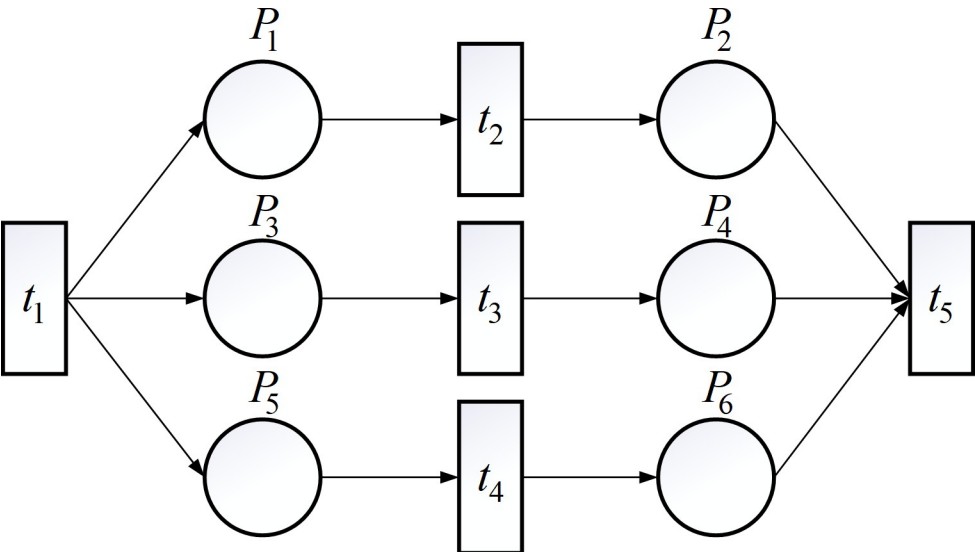

**Fig 2. Petri network concurrency relationship description.**

TTPN link concurrency logic description: if there are two or more transitions in the enabling state at the same time, $\exists t_1, t_2$ meets $M_0[t_1>$, $M_0[t_2>$. The excitation of one transition does not affect the excitation of another transition. If $M_0[t_1>M_1$, the process will record as $M_1[t_2>$. If $M_0[t_2>M_2$, the process will record as $M_0[t_2>M_2$. For example, $t_2$ for passenger service operations, $t_3$ or luggage and cargo operations, $t_4$ for aircraft operations. The parallel relationship of the three is shown in Fig 2:

## 2.2 The turnaround process modeling

The TTPN model is established for the A320, B777 and other series of commonly used flights in a large airport in China, as shown in Fig 3. $t_1$: Landing, approach sliding, and wheel blocks blocking. $t_2$: Gallery bridge docking and deboarding. $t_3$: Cleaning. $t_4$: Catering. $t_5$: Refueling. $t_6$: Toilet servicing and potable water servicing. $t_7$: Boarding. $t_8$: Passenger and cargo synchronization. $t_9$: Unloading preparation work. $t_{10}$: Unloading, loading, and closing cargo doors. $t_{11}$: Gallery bridge evacuating, wheel blocks removing and tractor operation. $t_{12}$: Departure sliding.

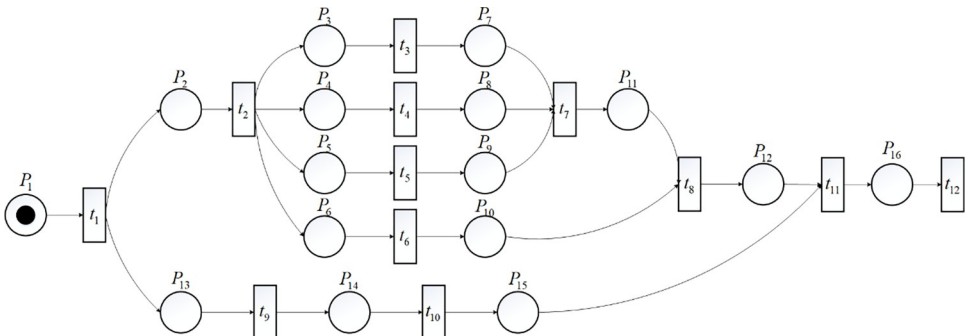

**Fig 3. Petri network description of flight turnaround operation process.**

## 3. Dynamic prediction of turnaround time

Because the information faced by the system in deciding whether the flight will be delayed is incomplete, how to predict the probability in the historical data information field is particularly important. There are uncertainties in task execution time and randomness in values among links of flights. However, there are generally connections in actual data. In this paper, Bayesian theorem is used to establish connections among links, which has become an important parameter for dynamic prediction of turnaround time.

### 3.1 Study on the parameters

Bayesian theorem is founded by Thomas Bayes, a British mathematician. It systematically describes the relationship between random variables, applies the observed phenomena to make subjective judgments on the relevant probability distribution, and then corrects them. It is used to reason about the probability model with causality. Uncertainty reasoning makes the model logically clear.

The TTPN can be defined as $(t_1, t\rightarrow)$, $t_1$ is the initial time. The TTPN contains the departure conditional probability distribution at the initial time $P(t_{i+1}|t_i)$. Formula 2 shows that when $t_i$ event occurs, the probability of causing $t_{i+1}$ event to occur is calculated to find out what is most likely to cause it to occur.

$$P(t_{i+1}|t_i) = \frac{P(t_{i+1})P(t_i|t_{i+1})}{\sum_{j=i+1}^{n} P(t_j)P(t_i|t_j)} \tag{2}$$

Where, $(t_1, t\rightarrow)$: the evolution mechanism of departure process. $P(t\rightarrow|t_1)$: The distribution between variables after the process has been updated.

According to the TTPN structure of the aircraft turnaround process, the conditional probability density of the turnaround time prediction time can be obtained according to the chain rule.

$$f(t_1, t_2, \ldots t_{12}) = f(t_{12}|t_{11}, \ldots t_1) \ldots f(t_2|t_1)f(t_1) \tag{3}$$

The conditional probability density of each link needs to be obtained in advance. Based on the classification of historical data, the prior probability density of each link and the joint distribution law relationship between each link are obtained.

The number of seats of different types of aircraft will be different, resulting in the impact of turnaround time. Combined with the Civil Aviation Administration's minimum turnaround time standard Table 1, the flights studied in this paper are divided into medium-sized flights (C) and large flights(D).

**Table 1. Flight classification standard and its minimum turnaround time.**

| Category | Seating | Representative models | Airports with a passenger throughput of more than 30 million passengers (including) (min) |
|---|---|---|---|
| A | 0~60 | MA60、EMB145、ATR72 | 45 |
| B | 61~150 | B733、B737-600、A319、CRJ7 | 55 |
| C | 151~250 | B763、B788、B738、A310、A320、A321、C919 | 65 |
| D | 251~500 | B747、B777、A300、A306、A330、MD11 | 75 |
| E | >500 | A388、A380 | 120 |

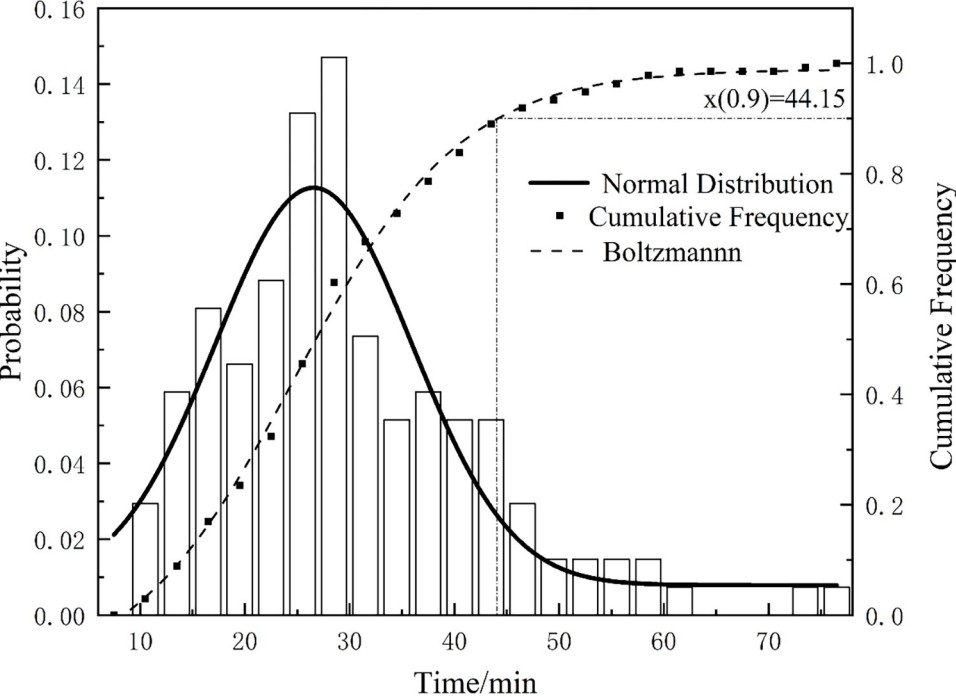

**Fig 4. The prior probability distribution of D-type catering link.**

Fig 4 shows the prior probability distribution fitting of the D-type flight catering link. Fig 5 reflects the joint distribution law of the catering and boarding links. It can be seen from Fig 4 that the normal fitting distribution can largely reflect the actual situation.

Using the Shapiro-Wilke test method to determine whether the normal fitting curve can truly reflect the historical data, the specific steps are as follows:

1. Arrange according to the sample data from small to large $(X_1, X_2, X_3, \ldots X_n)$.

2. Find out the $a_{in}$ values corresponding to $n$ in the Shapiro-Wilke coefficient table.

3. The statistic of this test is:

4. Determine the test level $\alpha$. Find out the $W(n,\alpha)$ value. When $W > W(n,\alpha)$, it is considered that the population conforms to the normal distribution, that is, the null hypothesis is accepted.

$$W = \frac{\left[\sum_i a_{in}\left(X_{(n+i-1)} - X_{(i)}\right)\right]^2}{\sum_{i=1}^n \left(X_{(i)} - \bar{X}\right)^2} \tag{4}$$

Where, $x_{(i)}$: The $i$th minimum number in the sample. $\bar{X}$: Average number of sample data.

Where the numerator $\sum_i$, when it is even, it is $\sum_{i=1}^{\frac{n}{2}}$. When it is odd, it is $\sum_{i=1}^{\frac{n+1}{2}}$.

## 3.2 Dynamic prediction steps of turnaround time

Because the prediction of the future is difficult to be consistent with the previous situation in many cases, for most normal flights, the delay decision obtained from the prediction results

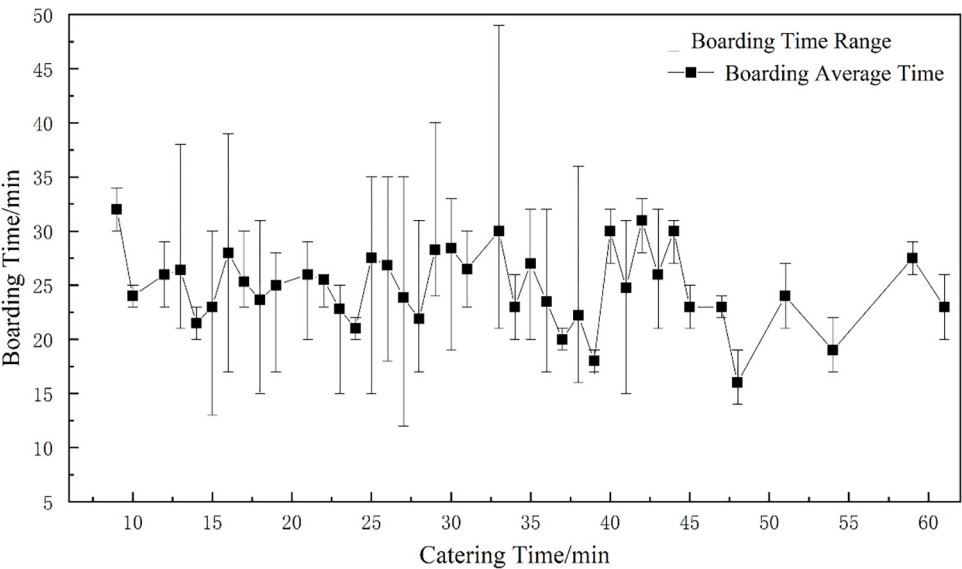

**Fig 5. The joint distribution law of D-type flight catering-boarding.**

often has a great risk, and the risk is closely related to the probability. The data with a cumulative frequency of less than 90% are included in the series. The dynamic prediction steps of turnaround time are shown in Fig 6.

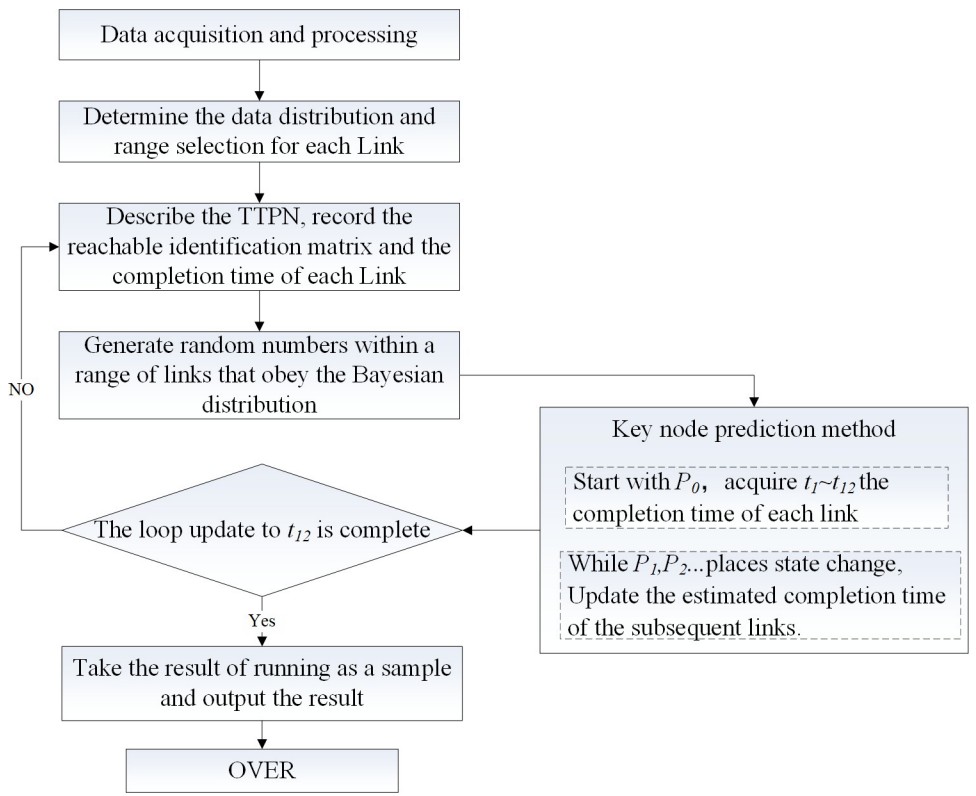

**Fig 6. Dynamic prediction steps of turnaround time.**

Step 1: According to the operation service links represented by each transition in TTPN, the prior probability distribution normal fitting of each link is carried out by using historical data, and the joint distribution law between the related links is obtained.

Step 2: The Shapiro-Wilke test is used to confirm whether the prior probability normal fitting is valid, and the conditional probability distribution function is determined.

Step 3: Take the maximum probability of each link as the initial value, combined with Monte Carlo simulation to determine the range of conditional probability for each round;

Step 4: According to the logical relationship of the operation service link in TTPN, the Monte Carlo simulation value of each round is used as the duration of the subsequent link change, and the order of the link is updated.

Step 5: With the continuous progress of each process, repeat the prediction of the completion time of the follow-up link. If the simulation order is less than 12, go to step 3.

Step 6: According to the results of the test set and training set, the prediction level is evaluated by calculating the root-mean-square error and the mean absolute error.

$$RMSE(S, \wedge S) = \sqrt{\frac{1}{n}\sum_{t=1}^{n}\left(s_t - \wedge s_t\right)^2} \tag{5}$$

$$MAE(S, \wedge S) = \frac{1}{n}\sum_{t=1}^{n}(s_t - \wedge s_t) \tag{6}$$

## 4. Results

This article uses the actual effective operational data of a certain airline at an airport in North China in 2023. The statistical objects include non-overnight flights near the airport.

### 4.1 Data processing

The information involved in 12 links is classified and processed. Some of the results are shown in Tables 2 and 3, which are used as parameters for dynamic prediction of each link. It is assumed that the duration of each link is only affected by the maximum value in the link closely connected with him.

### 4.2 Flight verification analysis

The training analysis of the D-type flights data on a certain day is shown in Fig 7. The verification set is distributed around $y = x$, and the goodness of fit $R^2 = 0.608$ can reflect the historical data.

**Table 2. The prior probability distribution parameters of the part of the link (min).**

| Links | C-type | | D-type | |
|---|---|---|---|---|
| | AVE | STD | AVE | STD |
| $t_1$ | 8.85 | 1.90 | 11.24 | 1.80 |
| $t_4$ | 25.17 | 12.09 | 25.21 | 8.31 |
| $t_3$ | 19.98 | 6.50 | 22.05 | 8.99 |
| $t_5$ | 19.84 | 1.23 | 18.04 | 4.09 |
| $t_7$ | 25.35 | 7.25 | 25.16 | 6.65 |

**Table 3. The joint distribution law of partial $t_4$ – $t_7$ for D-type flights (min).**

| $t_4$ | $t_7$ | $t_4$ | $t_7$ | $t_4$ | $t_7$ |
|---|---|---|---|---|---|
| 17 | [23,30] | 24 | [20,22] | 30 | [19,33] |
| 18 | [15,31] | 25 | [15,35] | 31 | [23,30] |
| 19 | [17,28] | 26 | [18,35] | 33 | [21,49] |
| 21 | [20,29] | 27 | [12,35] | 34 | [20,26] |
| 22 | [23,26] | 28 | [17,31] | 35 | [20,32] |
| 23 | [15,25] | 29 | [24,40] | 36 | [17,32] |

An A333 airliner on a certain day is selected for dynamic verification. It is known that the number of seats on the flight is 294, which belongs to the D-type flight. The actual transit time of the flight is 117 minutes. The dynamic prediction process and results of each link are shown in Table 4.

20 D-type flights were randomly selected for simulation, and the dynamic predicted turnaround time value and actual value of each flight were obtained as shown in Fig 8. As the process progresses, the RMSE between the predicted value and the actual value is 3.75 min, and the MAE is 3.40 min.

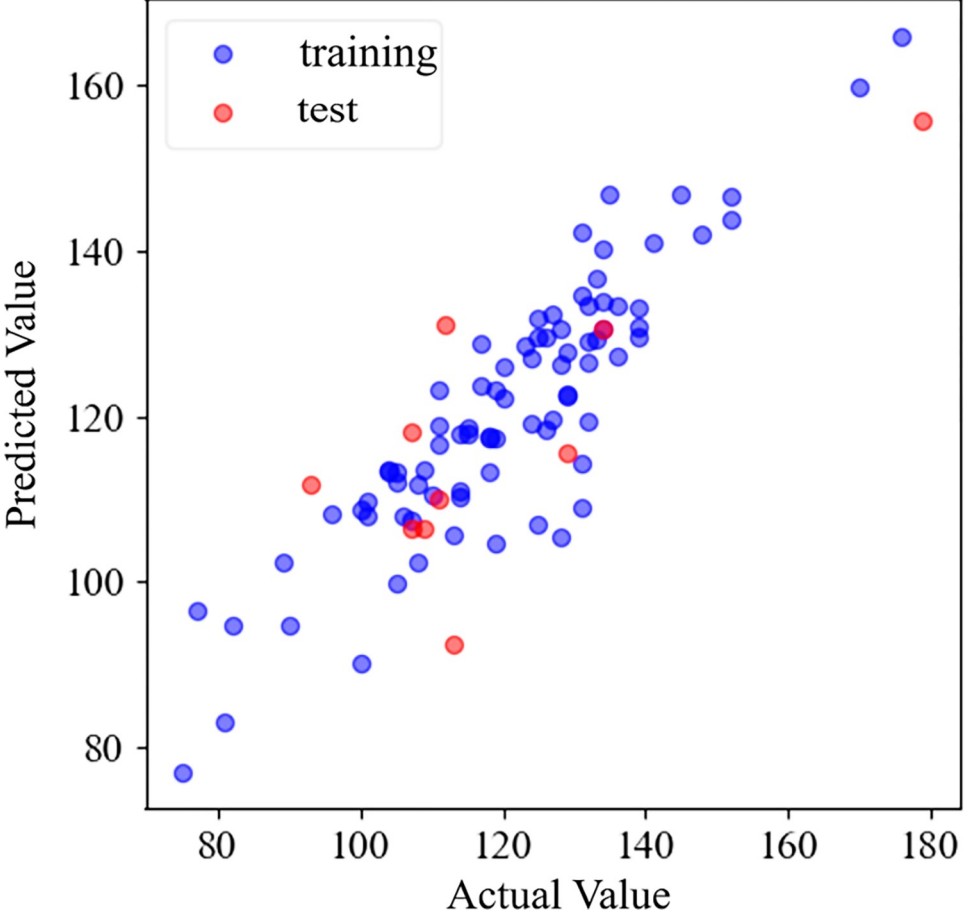

**Fig 7.** Data training situation.

**Table 4. Dynamic prediction of the process flow and results of an A333 airliner (min).**

| $t_1$ | $t_2$ | $t_9$ | $t_6$ | $t_5$ | $t_4$ | $t_3$ | $t_7$ | $t_8$ | $t_{10}$ | $t_{11}$ | $t_{12}$ |
|---|---|---|---|---|---|---|---|---|---|---|---|
| 11 | 15 | 27 | 35 | 33 | 40 | 37 | 65 | 67 | 107 | 112 | 123 |
| 10 | 14 | 18 | 43 | 33 | 44 | 36 | 71 | 71 | 98 | 102 | 120 |
| 10 | 14 | 14 | 33 | 58 | 29 | 40 | 93 | 93 | 75 | 97 | 106 |
| 10 | 14 | 14 | 27 | 21 | 42 | 28 | 63 | 69 | 75 | 78 | 91 |
| 10 | 14 | 14 | 27 | 30 | 55 | 31 | 86 | 86 | 87 | 91 | 107 |
| 10 | 14 | 14 | 27 | 36 | 44 | 31 | 70 | 70 | 109 | 112 | 124 |
| 10 | 14 | 14 | 27 | 36 | 49 | 31 | 78 | 85 | 75 | 91 | 101 |
| 10 | 14 | 14 | 27 | 36 | 49 | 53 | 82 | 82 | 101 | 106 | 118 |
| 10 | 14 | 14 | 27 | 36 | 49 | 53 | 73 | 75 | 77 | 81 | 92 |
| 10 | 14 | 14 | 27 | 36 | 49 | 53 | 73 | 80 | 76 | 84 | 102 |
| 10 | 14 | 14 | 27 | 36 | 49 | 53 | 73 | 80 | 100 | 105 | 127 |
| 10 | 14 | 14 | 27 | 36 | 49 | 53 | 73 | 80 | 100 | 105 | 115 |

## 4.3 Evaluate the prediction level

The process data of the above 20 randomly selected D-type flights are output, and the actual time of each link is analyzed. The RMSE and the MAE are used to measure the prediction level of each link. The results are shown in Table 5.

It can be seen from Table 5 that the RMSE of cleaning, catering, refueling and cargo service is slightly larger, which is due to the wide distribution of operation time in these links. Combined with real-time data dynamic update, the predicted turnaround time can greatly reduce the error caused by these links.

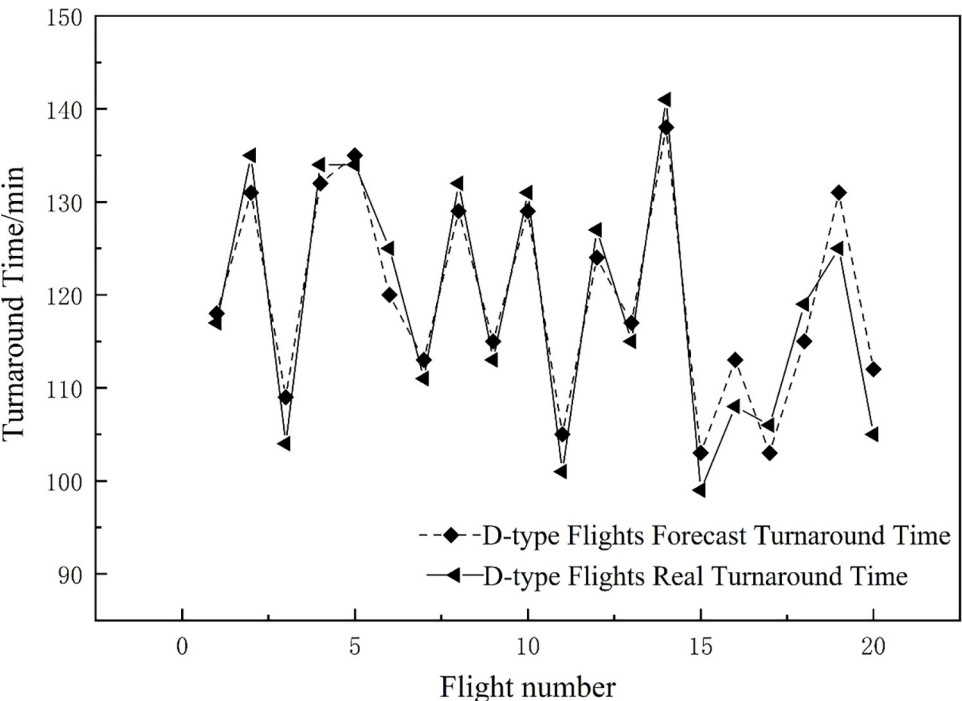

**Fig 8. The predicted value and actual value of some flight turnaround time.**

**Table 5. The comparison between the predicted value and the actual value of each link (min).**

| Link | RMSE | MAE | Link | RMSE | MAE |
|------|------|------|------|------|------|
| $t_1$ | 2.85 | 2.42 | $t_7$ | 3.36 | 3.00 |
| $t_2$ | 0.93 | 0.86 | $t_8$ | 2.42 | 1.29 |
| $t_3$ | 5.10 | 5.14 | $t_9$ | 1.40 | 0.85 |
| $t_4$ | 5.35 | 4.86 | $t_{10}$ | 6.55 | 5.71 |
| $t_5$ | 6.15 | 5.57 | $t_{11}$ | 1.13 | 0.71 |
| $t_6$ | 4.84 | 4.00 | $t_{12}$ | 2.78 | 2.29 |

## 5. Conclusions

In order to improve the accuracy of predicting flight turnaround time, this paper establishes a flight turnaround model based on TTPN by analyzing the flight turnaround operation process, and obtains the dynamic prediction method of flight turnaround time by combining Bayesian theorem. This method is used to predict the turnaround time of a large airport flight. Compared with the real value, the predicted the value of root-mean-square error is 3.75min and the value of average absolute error is 3.40min, providing a reliable research method for accurately estimating the flight turnaround time.

The research scope of this paper is the operation process that must be carried out in a stand for non-overnight single flights. The next step will be to study the stand environment, operation process execution time, flight busyness and other aspects to improve the accuracy and effectiveness of the turnaround time prediction model.

## Acknowledgments

Thanks for the guidance and technical support provided by the group. According to the relevant requirements and agreements of the scientific research project on which the article is based, the data not provided in the manuscript will not be disclosed.

## Author Contributions

**Conceptualization:** Yanyu Cui, Qingmiao Ding.

**Formal analysis:** Linyan Ma.

**Funding acquisition:** Yanyu Cui.

**Investigation:** Xuan He.

**Methodology:** Qingmiao Ding.

**Software:** Xuan He, Fanghui Xiao.

**Validation:** Bin Cheng.

**Writing – original draft:** Linyan Ma.

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
