## [Decision Letter · Decision Letter 0]

5 Apr 2024

PONE-D-24-05331Aircraft Turnaround Time Dynamic Prediction Based on Time Transition Petri NetPLOS ONE

Dear Dr. Ding,

Thank you for submitting your manuscript to PLOS ONE. After careful consideration, we feel that it has merit but does not fully meet PLOS ONE’s publication criteria as it currently stands. Therefore, we invite you to submit a revised version of the manuscript that addresses the points raised during the review process.

We look forward to receiving your revised manuscript.

Kind regards,

Poowin Bunyavejchewin

Academic Editor

PLOS ONE

Journal Requirements:

"the National Key Research and Development Project (2021YFB1600500-2021YFB1600502)."

5. We note that your Data Availability Statement is currently as follows: All relevant data are within the manuscript and its Supporting Information files.

**Additional Editor Comments:**

**ACADEMIC EDITOR:  **

 Please carefully review the comments and recommendations provided by the two reviewers and make the necessary revisions accordingly.==============================

Reviewers' comments:

Reviewer's Responses to Questions

**Comments to the Author**

1. Is the manuscript technically sound, and do the data support the conclusions?

Reviewer #1: Yes

Reviewer #2: Yes

2. Has the statistical analysis been performed appropriately and rigorously? 

Reviewer #1: Yes

Reviewer #2: Yes

3. Have the authors made all data underlying the findings in their manuscript fully available?

Reviewer #1: Yes

Reviewer #2: Yes

4. Is the manuscript presented in an intelligible fashion and written in standard English?

Reviewer #1: Yes

Reviewer #2: Yes

5. Review Comments to the Author

Reviewer #1: Manuscript Number: PONE-D-24-05331

Manuscript Title: Aircraft Turnaround Time Dynamic Prediction Based on Time Transition Petri Net

The study “Aircraft Turnaround Time Dynamic Prediction Based on Time Transition Petri Net” is the study proposes a new reliable research method for accurately predicting aircraft turnaround time. The objective of the study is very interesting, and the result is very useful to daily work of airports operation. However, its need some major/minor revision to improve. Some suggestion are as follow:

Abstract:

1. Abstract is reflected the manuscript. However, order of some line should be revised.

2. The last line “This paper proposes …..turnaround time.” is similar explanation with second sentence “In this paper, by analyzing the aircraft… is purpose.” So, please revised as a one sentence. And the last line should be the useful of this study output or future research direction.

3. Line 24 to 25, Please avoided short form in abstract. Authors should write only the long form of RMS and MAE , E.g. “the root-mean-square error 3.75 minutes and the mean absolute error 3.40 minutes can be calculated.

Introduction

1. Figure 1, 2 and 6 should be revised to get clear resolution like other figures.

2. The study objective is explained and clearly, but the significance of this work and structure of manuscript should be write in last part of this section.

Material and Method

1. Authors should mention about data information in details before the explanation of methods. Not is result section.

2. What historical data is used for the study? By time or total number of events?

3. Authors should consider other impact on airport operation management in their calculation such as weather delay situation in future study.

Result and discussion

1. Figure 7 should be revised for clear resolution.

2. Please move two square error equations to method section. Result section should be only results of everything.

General Comment

1. Authors should mention the airports list or location map where the data are taken to get more reader interesting.

2. Please put the section number to identify the major section and revised some section titles, For E.g. “Evaluate the prediction level of each link” is like a sentence or figure caption, not a title. It should be only ‘Evaluate the prediction level”.

3. The study objective is proposed to new methodology for future operation study, thus please highlight the method procedure in manuscript. Please move minor result table to supplementary section. And if possible, revised these tables to figures to put in manuscript.

Reviewer #2: comments :

i. Add more applications related to proposed mathematical modeling.

ii. Add more physical explanation about the mathematical modeling.

iii. The paper should be prepared according to the instructions of this journal.

iv. The title of the manuscript should be more concise.

v. Author should justify what are the novel contributions of their study using "DTM" approach.

vi. The following studies would be useful for the readers:

a) https://doi.org/10.1007/s11771-023-5398-1

b) https://doi.org/10.1080/17455030.2022.2123115

c) https://doi.org/10.1142/S0217984921505527

d) https://doi.org/10.3390/math9172139

6. PLOS authors have the option to publish the peer review history of their article (what does this mean?). If published, this will include your full peer review and any attached files.

Reviewer #1: **Yes: **Kyaw Than Oo

Reviewer #2: No

---

## [Author Response · Author response to Decision Letter 0]

15 May 2024

Details in the Attach Files - Response to reviewers.docx

---

## [Decision Letter · Decision Letter 1]

28 May 2024

Aircraft Turnaround Time Dynamic Prediction Based on Time Transition Petri Net

PONE-D-24-05331R1

Dear Dr. Ding,

We’re pleased to inform you that your manuscript has been judged scientifically suitable for publication and will be formally accepted for publication once it meets all outstanding technical requirements.

Kind regards,

Poowin Bunyavejchewin

Academic Editor

PLOS ONE

Additional Editor Comments (optional):

Both reviewers recommended the same: to accept this revised version of the manuscript. I have no objections.

Reviewers' comments:

Reviewer's Responses to Questions

**Comments to the Author**

1. If the authors have adequately addressed your comments raised in a previous round of review and you feel that this manuscript is now acceptable for publication, you may indicate that here to bypass the “Comments to the Author” section, enter your conflict of interest statement in the “Confidential to Editor” section, and submit your "Accept" recommendation.

Reviewer #1: All comments have been addressed

Reviewer #2: All comments have been addressed

2. Is the manuscript technically sound, and do the data support the conclusions?

Reviewer #1: Yes

Reviewer #2: Yes

3. Has the statistical analysis been performed appropriately and rigorously? 

Reviewer #1: Yes

Reviewer #2: Yes

4. Have the authors made all data underlying the findings in their manuscript fully available?

Reviewer #1: Yes

Reviewer #2: Yes

5. Is the manuscript presented in an intelligible fashion and written in standard English?

Reviewer #1: Yes

Reviewer #2: Yes

6. Review Comments to the Author

Reviewer #1: The authors revised the manuscript well, and it is totally improved to published standard. It can be published now after some minor corrections.

1. Please add the short meaning of P and t in Fig. 2 and 3 captions.

Reviewer #2: agree to accept. the authors address all the comments

7. PLOS authors have the option to publish the peer review history of their article (what does this mean?). If published, this will include your full peer review and any attached files.

Reviewer #1: **Yes: **Kyaw Than Oo

Reviewer #2: No

---

## [Editor Report · Acceptance letter]

13 Jun 2024

PONE-D-24-05331R1 

PLOS ONE

Dear Dr. Ding, 

I'm pleased to inform you that your manuscript has been deemed suitable for publication in PLOS ONE. Congratulations! Your manuscript is now being handed over to our production team.

Kind regards, 

on behalf of

Mr. Poowin Bunyavejchewin 

Academic Editor

PLOS ONE